# Complex Networks and Interacting Particle Systems

**DOI:** 10.3390/e25111490

**Published:** 2023-10-27

**Authors:** Noam Abadi, Franco Ruzzenenti

**Affiliations:** Integrated Research on Energy, Environment and Society (IREES), Energy and Sustainability Research Institute Groningen (ESRIG), University of Groningen, Nijenborgh 6, 9747 AG Groningen, The Netherlands; f.ruzzenenti@rug.nl

**Keywords:** complex networks, interacting systems, maximum entropy, statistical physics, Lennard-Jones

## Abstract

Complex networks is a growing discipline aimed at understanding large interacting systems. One of its goals is to establish a relation between the interactions of a system and the networks structure that emerges. Taking a Lennard-Jones particle system as an example, we show that when interactions are governed by a potential, the notion of structure given by the physical arrangement of the interacting particles can be interpreted as a binary approximation to the interaction potential. This approximation simplifies the calculation of the partition function of the system and allows to study the stability of the interaction structure. We compare simulated results with those from the approximated partition function and show how the network and system perspective complement each other. With this, we draw a direct connection between the interactions of a molecular system and the network structure it forms and assess the degree to which it describes the system. We conclude by discussing the advantages and limitations of this method for weighted networks, as well as how this concept might be extended to more general systems.

## 1. Introduction

The field of network theory is a growing branch of science aimed at representing and understanding interacting systems. The main notion is that the system can be represented by listing its components, known as nodes, and the connections between each pair of components, called links [1,2]. These simple requirements allow for the description and common study of a wide range of phenomena, generalized by studying the structural properties defined on the nodes and links of the network representation instead of on the system itself [3]. Structural properties range from simply the number of nodes and links to fractal dimensions related to notions of space occupied by the network [4].

Interacting particle systems serve, in many senses, as a reference case for network concepts [5] due to the natural representation that they enjoy as networks (in the form of nodes that represent particles connected by edges that represent a stable equilibrium distance) [6,7,8]. This representation inherently shares well-defined notions of nodes (particles) and an intrinsic relation between interactions and structure (e.g., molecular forces mediated by physical distance) grounded on physical concepts and experimental evidence. It has seen applications ranging from the study of molecule recognition [9] to self-organization [10] and to the notion of complexity [11,12]. In the field of complex networks theory, it has become increasingly clear that a better understanding of the relation between network structure and the interactions of the system would be useful to both the application and the framework [13,14]. Additionally, network structure formation models often draw on the notion of structure through link creation and destruction [15,16], which is natural to molecular systems in the same sense that the network representation of the spatial configuration is, and it provides a very intuitive view of the formation process. This motivates a positive outlook for complex networks providing a general view on patterns observed between interacting systems at different scales [17].

On the other hand, within fixed scales (in particular, at the molecular scale), network analyses of structure tend to either take data-analytic approaches [18,19] or impose external “control parameters” [8] built on the notion of adaptive links [20]. In this context, a change in temperature which allows certain molecular structures to emerge would be modeled by setting an external parameter which regulates how the links are created and destroyed. The problem with this is that the emergent phenomenon (in this case, simply stable links in the network) is imposed instead of actually obtained from the interaction. A question that then naturally arises is whether it is possible to construct an “environment-independent” network model of the particle system which does not consider the links as adaptive and still learns something about structure formation.

In this work, we study the emergence of a molecular structure in a simulated Lennard-Jones particle system in thermodynamic equilibrium by representing it as a complex network. In particular, we remove the standard restriction of molecular graph representations depicting only stable links and interpret these as binary approximations of the interaction potential. The goal of this is to free the network representation of details beyond the interaction itself, as these can change according to the environment or definitions of stability. We study the structural properties of the resulting network in equilibrium at different temperatures and pressures and show how the binary approximation of the potential allows the use of network and thermodynamic properties in combination, in particular to detect where the studied network structure is stable in thermodynamic equilibrium.

Particle systems in thermodynamic equilibrium are subject to description by maximum entropy. One of the main results [21] is that such systems can be described by a partition function, which is essentially a function Z of a parameter β from which average values (such as the average kinetic and potential energy) in equilibrium can be calculated. For a system of particles 1≤i≤N with kinetic energy $mv_{i}^{2}/2$ and potential energy $V_{ij}\left(r_{i},r_{j}\right)$. The canonical partition function is calculated as
(1)$$Z\left(\beta \right)=\int \exp\left(-\beta \left(mv_{i}^{2}/2+\sum_{i}\sum_{j>i}V_{ij}\left(r_{i},r_{j}\right)\right)\right)\Pi_{i=1}^{N}dv_{i}dr_{i}\;.$$
As the potential energy is independent of the velocity, the integrations over velocities can be separated from the spatial term. The former gives the partition function of an *N* classical particle system with no interactions, $Z_{v}={\left(\int \exp\left(-\beta mv^{2}/2\right)dv\right)}^{N}$. Assuming unit mass and two dimensions, this will result in a (kinetic) energy K=1/β with β=1/T representing the temperature of a noninteracting gas with that amount of (kinetic) energy, and the corresponding distribution of velocities will be Maxwell–Boltzmann. However, the spatial part is, in general, very difficult to calculate. We show that the network representation introduced in this work simplifies this task. Finally, we remark that while maximum entropy is well-established for thermodynamic equilibrium, the field of maximum entropy production [22,23] is a developing counterpart for out-of-equilibrium systems through which our approach might be extended.

## 2. Methods and Experiments

### 2.1. Lennard-Jones Potential

The Lennard-Jones potential is a useful, well-studied idealization of the interactions between two atoms at a molecular scale. In this work, the potential is written as
(2)$$\begin{array}{cc} V(r) & =V_{o}\left[{\left(\frac{r_{o}}{r}\right)}^{12}-2{\left(\frac{r_{o}}{r}\right)}^{6}\right]\\ \Rightarrow \vec{F}\left(r\right)=-\vec{\nabla }V\left(r\right) & =12\frac{V_{o}}{r_{o}}\left[{\left(\frac{r_{o}}{r}\right)}^{13}-{\left(\frac{r_{o}}{r}\right)}^{7}\right]\hat{r}\;. \end{array}$$

For two particles, a short distance $r<r_{o}$ between them produces an enormous repulsive force. At long distances $r>r_{o}$, it is very weakly attractive, and in the middle lies the equilibrium point $r=r_{o}$, as shown in Figure 1. This allows for particles with low kinetic energy (below $V_{o}$) to “bind” together. If the kinetic energy is much larger than $V_{o}$, the energy decrease close to $r_{o}$ is practically unnoticed, but a minimum distance is eventually reached at which the repulsion forces win over and the particles bounce back away from each other. These two limits represent low and high temperature, respectively, and between them lie a diversity of possibilities which give rise to different network structures.

For the particle simulation, we took the Lennard-Jones potential parameters to be $V_{o}=1$ and $r_{o}=1$. N=900 particles of mass 1 were set on a uniform 2-dimensional grid with cells of side *d* with random velocities, uniformly distributed between $-V_{M}$ and $V_{M}$. The total force on each particle due to its interactions with others allows for the numerical integration of the equations of motion of each particle using a leap-frog algorithm [24]. Initial conditions varied were *d* (between 1 and 4), in order to control initial potential energy through particle separation, and $V_{M}$ (between 0.1 and 5), in order to control initial kinetic energy. The size of time steps and total simulation time were chosen in order to ensure energy conservation and at least 3/4 of simulation time in equilibrium (constant proportions of kinetic and potential energy). Additionally, a set of “cooled” conditions were obtained, continuing the evolution of the last step of simulations started with V=0.1 after multiplying their velocities at this step by 0.5 (effectively cooling the system). Once these reached equilibrium and stayed there for at least 3/4 of the simulation time, the process was repeated with the last step of the new simulation. This was performed four times for each initial separation.

### 2.2. Network Construction and Stability

As discussed in the introduction, we aim to construct a network representation that captures the interactions between the particles, without relying on the particular notion of equilibrium (which in this case is thermodynamic). For this, we first consider an approximation of the interaction potential as a step function of unspecified heights $V_{a}$ and $V_{b}$, with the step $\Delta V=V_{a}-V_{b}$ at a separation radius $r_{T}$, that is, $V(|r_{i}-r_{j}\left|\right)\approx V_{a}\Theta (r_{T}-|r_{i}-r_{j}\left|\right)+V_{b}\Theta (|r_{i}-r_{j}\left|-r_{T}\right)$. One might protest that this is not a suitable representation, as it does not allow for repulsive potentials. However, we argue that one can imagine that a real collision is replaced by a virtual information exchange, where the particles exchange their properties, as shown in Figure 2. This replacement is valid for the regions far away from the interaction range (i.e., it is valid where the particle trajectories are approximately straight lines, which in a two-level potential would be the outer range $|r_{i}-r_{j}|>r_{T}$), while for the short range, it is equivalent to projecting the particle velocities onto their initial unitized velocity vectors.

With the approximation, the inner radius is then considered as the “active” region of the interaction between different particles *i* and *j*, defining a binary undirected network
(3)$$A_{ij}=\left(1-\delta_{ij}\right)\Theta (r_{T}-|r_{i}-r_{j}\left|\right)\;.$$
From it, we study the degree $k_{i}$, which (in the two-level approximation of the potential) represents the number of particles with which *i* is interacting with
(4)$$k_{i}:=\sum_{j}A_{ij}\;,$$
assortativity *a*, the correlation between the number of neighbors of a randomly selected pair of connected nodes,
(5)$$\begin{array}{cc} L & =\sum_{i}k_{i}\\ \alpha & =\sum_{i}k_{i}^{2}/L\\ a: & =\frac{\sum_{i}k_{i}\sum_{j}A_{ij}k_{j}/L-\alpha^{2}}{\sum_{i}k_{i}^{3}/L-\alpha^{2}} \end{array}$$
and clustering coefficient $c_{i}$, for the tendency of pairs of links to form triangles,
(6)$$c_{i}=A_{ii}^{3}/\left(k_{i}\left(k_{i}-1\right)\right)$$
as they represent the first three orders of network structure description (i.e., first, second and third neighbors).

Note that the links are not only independent of the particular notion of equilibrium but also of $V_{a}$ and $V_{b}$. Instead of fixing these values into the network model, they are inferred from a combination of network and system data (in our case, the simulations). This will allow the same binary representation of the interactions to replicate physical system properties as temperature changes and, fundamentally, the differentiation of stable ($V_{a}<V_{b}$) and unstable ($V_{a}>V_{b}$) regimes of the network structure. The value of $r_{T}$, on the other hand, must be chosen somewhat arbitrarily. We take the value $r_{T}=r_{o}=1$ as it is the minimum of the potential. Consider that, otherwise, a particle starting with arbitrarily low energy will produce a “discontinuity” in the representation if the energy of the system is increased enough for it to cross the value of the potential energy at some other $r_{T}$ which does not correspond to the minimum. This may be an interesting property in some contexts but not for the purpose of this study.

We show two different ways we can calculate $V_{a}$ and $V_{b}$. In the first, we study the high-temperature limit by assuming that “free” pairs of particles (those at a distance greater than $r_{T}$) have interaction potential $V_{b}=0$ and calculating $V_{a}$ from the average potential energy measured from the simulations. In the second, we do not impose either $V_{a}$ or $V_{b}$, obtaining both from the potential and number of links measured in the network. The goal of these two approaches is to show the flexibility and limitations of the method to physical assumptions we may make. In both cases, the relation between the calculated properties and the ones measured from the system are obtained by using the partition function of the approximated potential, which is obtained in Appendix A. In Appendix B, we discuss how this method can be extended to other potentials and weighted networks.

## 3. Results and Discussion

As the gas is not initially in equilibrium (its velocities are distributed uniformly), the simulations show a dynamic phase before reaching it. This means that initial kinetic and potential energies do not correspond to equilibrium ones. In Figure 3, we show the (absolute value of) equilibrium proportions of potential to kinetic energy as a function of the initial velocity for the different initial cell sides used. The ratios range from around 10 times as much potential to kinetic energy to the inverse of that, showing that a wide variety of conditions are covered.

All the results presented in this work are in equilibrium conditions. As simulation parameters were chosen in order to obtain at least 3/4 of the simulation time in equilibrium, we use the final half of the simulation (instead of 3/4, as the transition from dynamics to equilibrium is not a sudden event) when we refer to this regime.

### 3.1. Temperature and Density

As discussed in the introduction (and in much more detail by Kardar [21]), the distribution of particle speeds $v=|\vec{v}|$ in equilibrium is the same as that of a noninteracting particle system with the same kinetic energy; so, it can be used to determine the temperature *T* by fitting the two-dimensional Maxwell–Boltzmann distribution. In two dimensions, this temperature is also equal to the average kinetic energy, which gives us a way to verify the fitted temperature and whether the simulation has reached equilibrium.
(7)$$\rho \left(v\right)=\frac{v}{T}\exp\left(-\frac{v^{2}}{2T}\right)$$
On the left hand side of Figure 4, we show the temperatures as a function of the initial velocities for the different initial particle separations, obtained from adjusting the temperature *T* in Equation (Equation 7) to the velocity distribution measured from the simulations. On the right hand side, we show the agreement between the inverse of the temperature with the kinetic energy.

As for the spatial distribution, the average separation of two particles is given by the average of the average distance from each particle to others,
(8)$$D^{2}:=\frac{1}{N}\sum_{i}\frac{1}{2(N-1)}\sum_{j>i}{\vec{r}}_{ij}^{2}=\frac{N}{N-1}\left(\langle {\vec{r}}^{2}\rangle -{\langle \vec{r}\rangle }^{2}\right)\;.$$
This average interparticle distance (squared) defines a cell of equal sides *s* whose area is the inverse of the particle density. Knowing that $D^{2}=2s^{2}$, the particle density is $\rho =2/D^{2}$. Its equilibrium value is shown as a function of the initial velocity in Figure 5. The error bars represent density fluctuations in the average distances of individual particles, $\Delta \rho =2/{\left(D^{2}\right)}^{2}\Delta D^{2}=2/{\left(D^{2}\right)}^{2}\mathrm{var}{\left(D^{2}\right)}^{1/2}$. The values of $D^{2}$ present a mostly constant density for high temperatures but an increasing trend below a certain temperature. The size of density fluctuations increases with density, suggesting that the spatial distribution is composed of dispersed clusters of tightly packed particles, and increasingly so the lower the temperature.

### 3.2. Network Properties

In Figure 6, we show the average degree of the network: on the left as a function of the temperature, while on the right as the potential energy with inverted sign. The degree of each node (particle) is calculated for each time step in equilibrium, and the average is taken over all time steps and nodes. All densities seem to present a two-phase behavior of average degree first decreasing with temperature (as the stable structure dissolves up to the phase transition) and then increasing once again (through random fluctuations of unstable interactions). As we see in Figure 5, particle separations are roughly constant at high temperatures, meaning that the increase in degree can only be attributed to the change in kinetic energy. This implies that slowly moving particles will reach others faster (increasing link formation), but due to the relatively low velocity, these will not be able to bounce off as easily (meaning that the corresponding increase in broken links is not as large). It is also interesting to note that in the high-potential (or low-temperature) regime, the degree becomes independent of the separation.

The distribution of degrees of different nodes is shown on the left-hand side of Figure 7. We see that it resembles a Gaussian distribution, suggesting that the degree is simply the result of identically distributed independent random variables. However, as the distribution is cut off at only six links (this is because the minimum potential energy is achieved in a uniform hexagonal grid), many concave functions will produce reasonable approximations. On the right-hand side, the average spectral density of a node’s degree over time is shown. The spectral density represents the “intensity” of fluctuations in different frequencies of a signal. This means that a slowly changing signal will have high spectral density at low frequencies (and low spectral density at high frequencies), while a rapidly changing one will have high spectral density at high frequencies and low spectral density at low frequencies. In this case, the signal is taken to be the degree of a single node as it varies over time in the simulation (for the equilibrium range), and the result shown in Figure 7 is the average of the resulting spectral densities over all nodes in the network.

In Figure 8, we show the assortativity of the network as a function of temperature. Assortativity represents the tendency of nodes to connect to those with a different or similar degree to themselves (between −1 and 1, respectively). The fact that assortativity is a global measure that we cannot average over nodes produces an image with considerably more noise than the other measures. However, we can still make out that assortativity is positive. This represents the fact that structure (as in connected particles) is always relatively similar throughout the system (it is unlikely to find dense agglomerations of particles with long chains extending from them, for example). Starting at low temperature, the particles form a hexagonal grid, so every node is identical (except for the boundaries). As temperature increases, some of the particles are freed from the grid and travel around, adding asymmetry to the grid and thus decreasing the assortativity. After a certain point, the free particles become the dominant property of the system (structure becomes unstable) and whatever is left of the grid is now the exception; so, as it is dissolved, nodes become more and more similar and assortativity increases again. This suggests a connection between assortativity minimum and the stability of a structure when there is a transition from repulsive to attractive behavior. It is then also interesting that the minimum of the degree happens simultaneously with assortativity, as can be seen in the degree–assortativity curves on the right of Figure 8.

In Figure 9, we show the clustering of the network as a function of the temperature and degree. The clustering coefficient represents the tendency of a link with two neighbors to form a closed cycle of three links. This is a useful structural property, as it measures how often pairs of connections “close”; so, the fact that it scales to the average degree independently of the density makes it interesting for models of structure formation.

### 3.3. Structural Stability

We now turn to calculating the parameters $V_{a}$ and $V_{b}$. Following Appendix A, we find that we can write the spatial term of the partition function
(9)$$Z_{r}={\left(L^{2}\exp^{-\beta V_{b}}\right)}^{N}\exp{\left(-\frac{\left(N-1\right)\pi r_{T}^{2}}{2L^{2}}\left(1-\exp^{-\beta \Delta V}\right)\right)}^{N}$$
where the first term corresponds to the contribution of “free” particle states with energy $V_{b}$ (which are beyond a distance $r_{T}$ to any other), while the second corresponds to the contributions from active interactions where the pair has energy $V_{a}=V_{b}+\Delta V$. The central assumption for this result to hold is that the interaction radii $r_{T}$ are small enough (with respect to the box size *L*) to assume that overlapping disks do so in pairs. The bad news is that this approach will likely fail for long-range potentials (nevertheless, long-range potentials are a problem in statistical physics), but the good news is that this is independent of the particular shape of the potential, as discussed in Appendix B. It is not clear how to modify the calculation in order to impose the cutoff at six links observed in Figure 7. In this form, there is a clear relation to the individual particle contribution $z_{1}$, just as for the partition function of the velocity. Since the order of integration in both the spatial and momentum space is subject to the same particle-renaming symmetry, which produces the Gibbs paradox if not accounted for [21], Equation (Equation 9) is technically missing a 1/N! factor. However, as we consider only fixed particle number cases, this is not expected to give us any problems. The interesting thing, however, is that any network representation will be subject to the same symmetry, which amounts to the row/column renaming of a network’s adjacency matrix. Studying entropy during the mixing of two initially independent networks in order to determine whether the Gibbs paradox also occurs is an interesting future line of research.

According to Equation (Equation 9), the average potential energy of the system is
(10)$$\langle V\rangle =-\partial_{\beta }\ln\left(Z\right)=NV_{b}+\frac{N\left(N-1\right)\pi r_{T}^{2}}{2L^{2}}\exp^{-\beta \Delta V}\Delta V$$
which must match the potential energy measured from the simulations to relate our network and physical model. The value of β is known, as it is given by the temperature (kinetic energy), but $V_{b}$ and ΔV are both free parameters. One of them will be determined by Equation (Equation 10), but the other is still unknown.

As a first approach, we simply assume the high-temperature limit in which the interactions beyond the strong repulsive force at short distances is negligible. In this case, the hard-shell approximation should hold, so we can set $V_{b}=0$. This leaves only $V_{a}=\Delta V$ to be determined numerically from Equation (Equation 10). Then, considering that all the potential energy is stored in links of energy ΔV, the number of links should be 〈V〉/ΔV. In Figure 10, we show, on the left-hand side, the height $V_{a}=\Delta V$ of the high-temperature limit ($V_{b}=0$) of the two-level potential. In the middle, the agreement between the measured and resulting potential energy shows that $V_{a}$ correctly recovers this value. On the right-hand side, we compare the number of links measured in the simulations with that expected from 〈V〉, using only data from simulations with T>1 to show the validity of the approximation beyond imposed values. It should be noted that the number of links varies much more in the low-temperature regime, making this value seem practically constant. Also, as the potential energy of the system is negative, so is ΔV, which seems to suggest that these links are stable. We interpret this as a failure of this approximation to provide results on the stability of structure due to the imposed value $V_{b}=0$.

For the second case, we do not impose either $V_{a}$ or $V_{b}$. We require an additional equation to obtain one of them. For this, we recall that a macrocanonical partition function $Z_{M}$ (where the average energy and number of components are fixed by the temperature β and the chemical potential μ) obeys
(11)$$-\partial_{\beta }\ln\left(Z_{M}\left(\beta ,\mu \right)\right)=\langle E\rangle -\mu \langle N\rangle$$
where 〈N〉 is the average value of a fluctuating number of components. We then propose that Equation (Equation 10) takes this form so that ΔV=−μ represents the chemical potential and $\langle N\rangle =N\langle k\rangle /2=N\left(N-1\right)\pi r_{T}^{2}\exp^{\beta \mu }/2L^{2}$ is the average number of links observed in the simulation. Note that μ tells us about the stability of the structure (μ>0 is stable while μ<0 is unstable) because of its definition from ΔV, which aligns with the notion of chemical potentials in statistical physics. In Figure 11, we show the values obtained for $V_{a}$ and $V_{b}$ in this approximation. We see that the structure becomes stable (μ<0→μ>0) below T=1. In Figure 12, we show the agreement between the number of links and the measured and calculated potential.

Finally, we must verify that this way of analyzing the stability of the structure agrees with the simulations. For this, consider that when the structure is unstable, links will form randomly throughout the nodes of the network. When the structure is stable, the links are persistently associated with a pair of nodes. In order to capture this, we first measure the average over time (in equilibrium) of the adjacency matrix of the network $A_{ij}=\sum_{t=0}^{\tau }A_{ij}\left(t\right)/\tau$. If the links are unstable, the whole matrix will have roughly the same value, while if they are stable, some pairs will have a high value and others a low one. The relative fluctuations around the mean value $\mathrm{var}\left(A_{ij}\right)/{\langle A_{ij}\rangle }^{2}$ (where both $\mathrm{var}\left(\cdot \right)$ and 〈·〉 are taken over links i,j) capture this effect, simultaneously correcting for the specific link density. In Figure 13, we compare this quantity (rescaled by a factor of 20 to be visible) to the chemical potential shown in Figure 11, showing that the latter predicts a transition from unstable to stable slightly before (when the temperature is being lowered) what is shown by the relative fluctuations.

## 4. Conclusions and Further Work

A Lennard-Jones particle system in thermodynamic equilibrium was mapped to a binary undirected network by using a two-level approximation for the interaction potential according to the distance between the particles. This was motivated by constructing a “environment-blind” (in reference only to the interactions, without resorting to emergent properties like the temperature) network model for particle interactions, which allowed us to simplify the calculation of the system partition function. Said partition function was shown to be able to reproduce the system energy and number of connections exactly, and its parameters were used to determine the stability of the interaction structure. The partition function can also be interpreted as a macrocanonical partition function with a varying number of links, defining a chemical potential which captures the same notion of structural stability. This supports the point of view of treating links in the network as the effective fluctuating “particles” presented in statistical mechanics of networks [25,26].

The specific network structural properties studied were the degree, assortativity and clustering coefficient, taken as representative of the first three orders of interaction in a network. All three properties exhibit a minimum as a function of the temperature and, in particular, clustering was found to have the same relation to degree across different densities. These minima are aligned with the change from a stable to unstable structure (middle- to high-temperature), suggesting the two are related. Indeed, as the binary network is essentially a partition into physically distinct regions of the interaction potential, with the physical properties of the system determined through the partition function of the approximation, then phase transitions that render certain structures stable correspond to changes in the “average behavior” of the system in each region. In our case, this is the transition from weak, short-range attractive forces in low temperature to strong repulsive (yet also short-range) interactions at high temperature. This suggests that under such changes in the structural properties, it may be better to increase the “resolution” of the network approximation, using a weighted network, which has not been tested and is interesting for future development in order to see if the method can improve accuracy. Another suggested way to perform this is to relate interaction terms beyond pairwise ones in the partition function to network properties, such as the degree distribution to estimate what order the interactions should be accounted for.

If we were to add a further approximation to the potential, for example, three ranges of distance instead of two, we would attain a better model of the potential but also a network with three possible connection values. It would require further conditions on the relation between these three values (beyond the potential energy and the number of links). Assuming that this is possible, increasing the number of values to a countable infinity would produce a weighted network with integer-valued levels, and finally, the continuous limit continuous weights that match the potential. This suggests that the “natural” network representation of a system that interacts through a potential is the potential itself. However, it comes at high analytical and computational difficulty, suggesting that other representations might be more convenient.

Independently of the most convenient representation, this draws a relation between a network link and the interaction energy contained in the region of space it represents. By checking whether these links are stable, we can tell if the energy is physically contained in a structure. In this case, the network (on average) resembles the traditional representation of particles at their stable distance. This method then allows to tell when energy is contained in structure, which could be a useful tool for a more systematic study of energy density across scales [27]. For this (and other applications beyond potential-mediated interacting particle systems), the reliance on maximum entropy is particularly convenient, as the conservation of energy, which puts kinetic and potential energy in the integrand of the partition function, can be replaced by a constraint on average values of arbitrary pairwise system states, which does not require the physical notion of an interaction potential. Any specific average value that is known to reproduce properties of a system through maximum entropy (for example, $\sum_{i,j}{\left(\ln\left(s_{i}\right)-\ln\left(s_{j}\right)\right)}^{2}/N^{2}$) then takes the place of the interaction potential, and the link represents the measure of this function (replacing the energy) contained in the states (replacing positions) corresponding to a link.

## Figures and Tables

**Figure 1 entropy-25-01490-f001:**
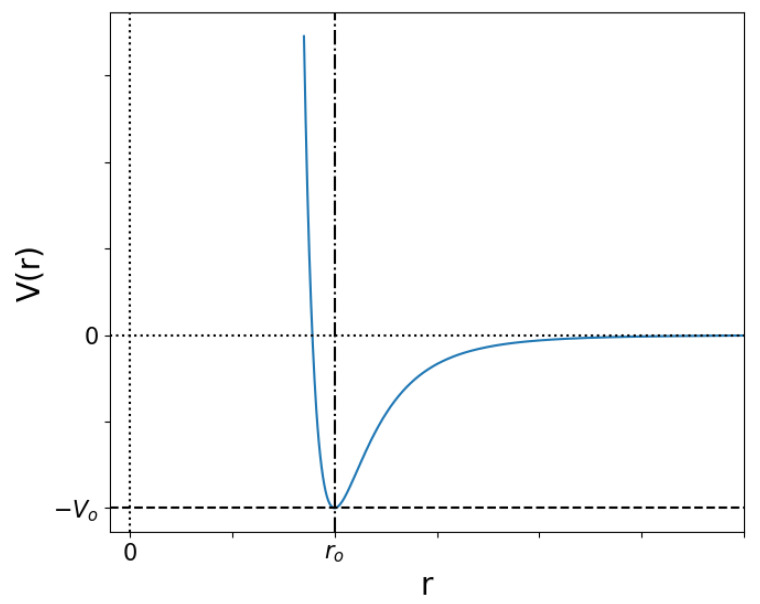
Potential energy as a function of distance in the Lennard-Jones potential in Equation (Equation 2). For kinetic energies below the minimum value $V_{o}$, particles will remain at a distance of approximately $r_{o}$.

**Figure 2 entropy-25-01490-f002:**
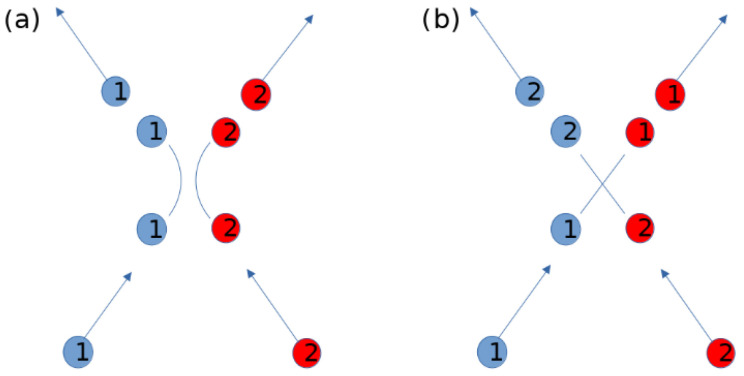
A repulsive interaction between two particles as shown in (**a**), can be replaced by an information exchange where each particle acquires the properties of the other, following straight trajectories, as shown in (**b**). We set a “name” (1 and 2) in order to track the particles in both scenarios, but the name is not a property as it is not exchanged.

**Figure 3 entropy-25-01490-f003:**
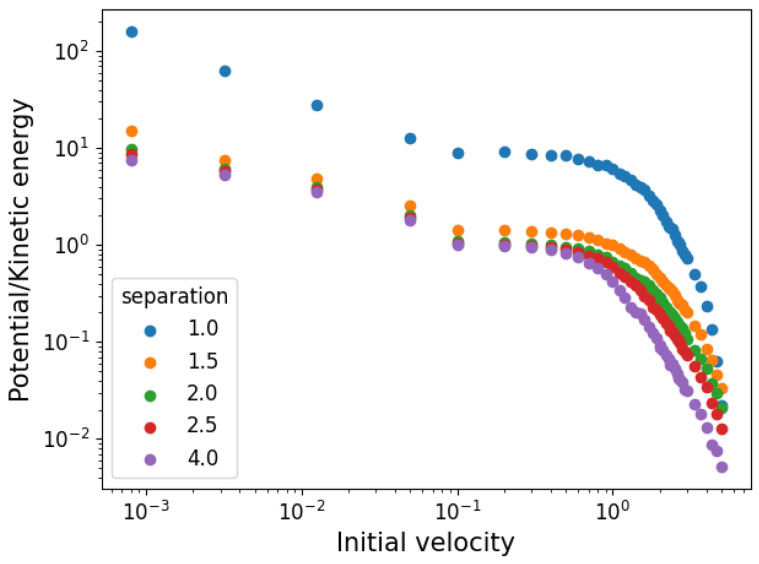
Ratio of equilibrium kinetic to potential energy as a function of initial velocity distribution boundaries.

**Figure 4 entropy-25-01490-f004:**
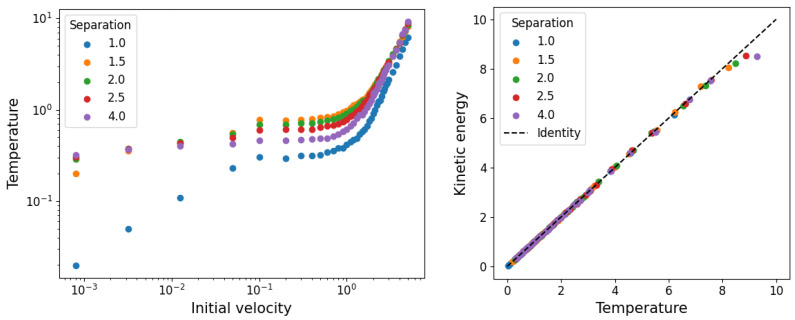
On the left: fitted temperature in equilibrium as a function of the initial velocity range $V_{M}$ for different initial separations. On the right: agreement between fitted temperature and the average kinetic energy.

**Figure 5 entropy-25-01490-f005:**
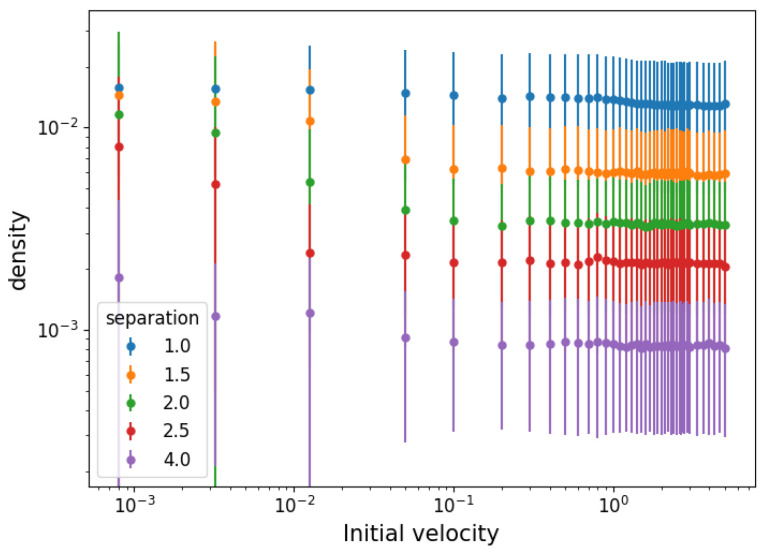
Density $\rho =2/D^{2}$ calculated from the particle separation *D* in equilibrium and fluctuations as a function of the initial velocity range $V_{M}$ for different initial separations.

**Figure 6 entropy-25-01490-f006:**
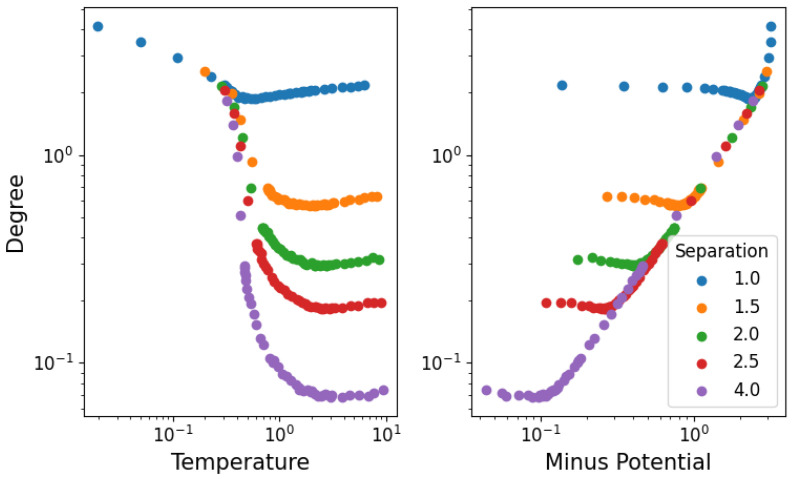
Average degree of nodes in equilibrium as a function of temperature for different densities, with error bars representing the spread of the degree values among nodes at any given time.

**Figure 7 entropy-25-01490-f007:**
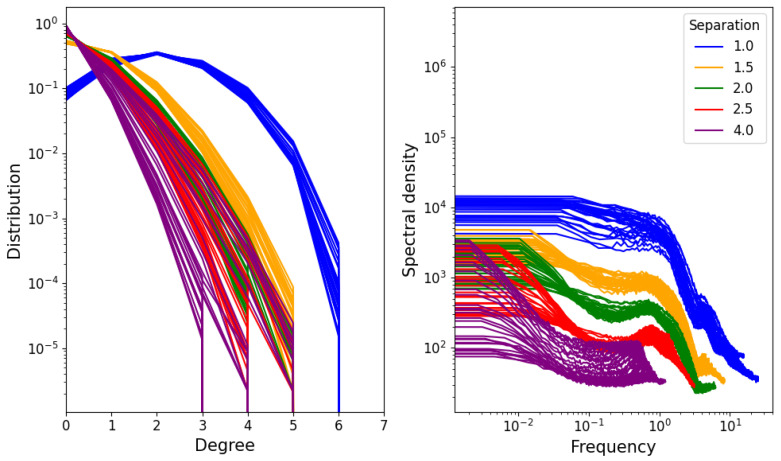
(**Left**) Degree distribution for different densities and temperatures. (**Right**) Average over nodes of degree spectral density.

**Figure 8 entropy-25-01490-f008:**
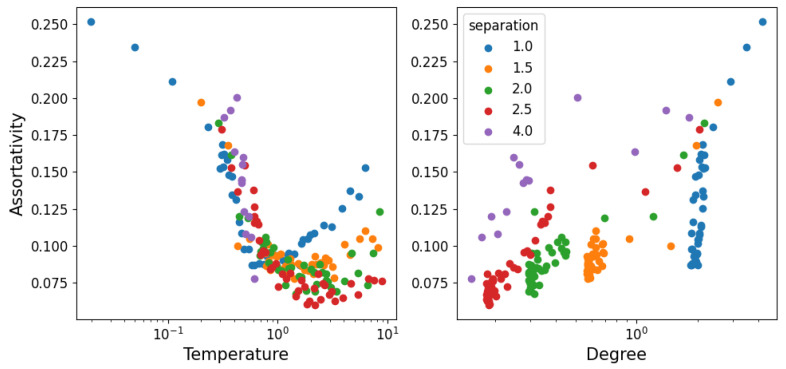
Assortativity (tendency of nodes to connect to others of similar degree) as a function of temperature (**left**) and degree (**right**). The figure has considerably more noise than the others, as assortativity is a global property, as opposed to the degree which we can average over nodes for each instant.

**Figure 9 entropy-25-01490-f009:**
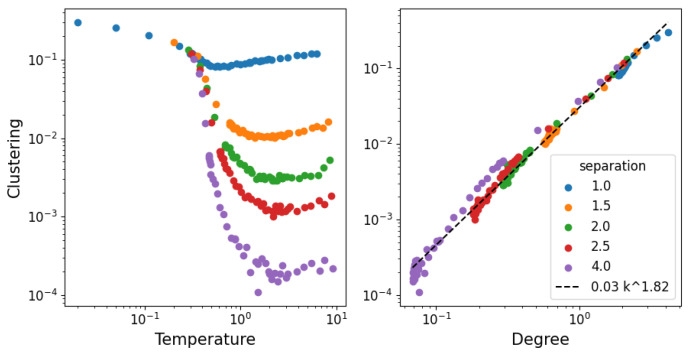
Clustering coefficient (tendency of pairs of connections to form triangles).

**Figure 10 entropy-25-01490-f010:**
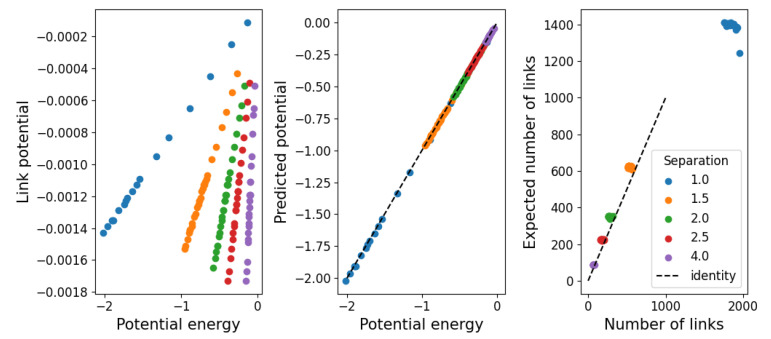
(**Left**) Potential $V_{a}=\Delta V$ determined from Equation (Equation 10) in the high-temperature limit $V_{b}=0$. (**Middle**) Agreement between measured and calculated potential. (**Right**) Number of links measured from the simulations in comparison with the number of links expected from 〈V〉/ΔV for temperatures T>1.

**Figure 11 entropy-25-01490-f011:**
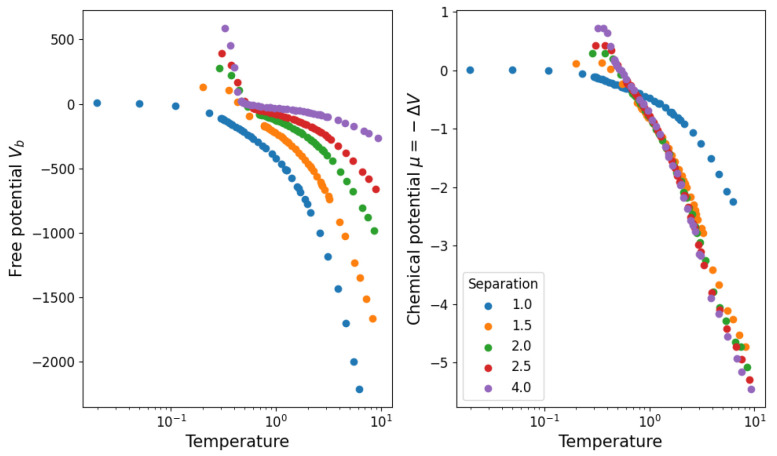
Partition function parameters calculated from the measured average potential energy and average number of links in the system.

**Figure 12 entropy-25-01490-f012:**
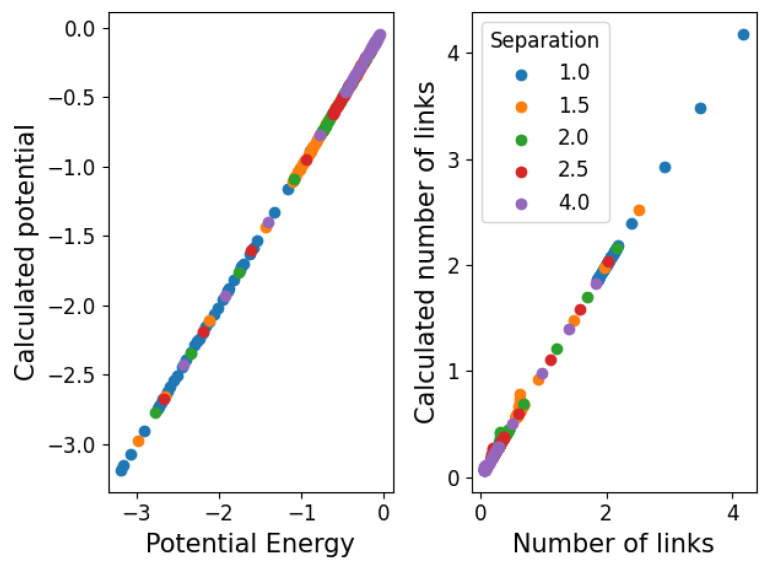
Agreement between simulated and calculated values of the potential energy and the number of links.

**Figure 13 entropy-25-01490-f013:**
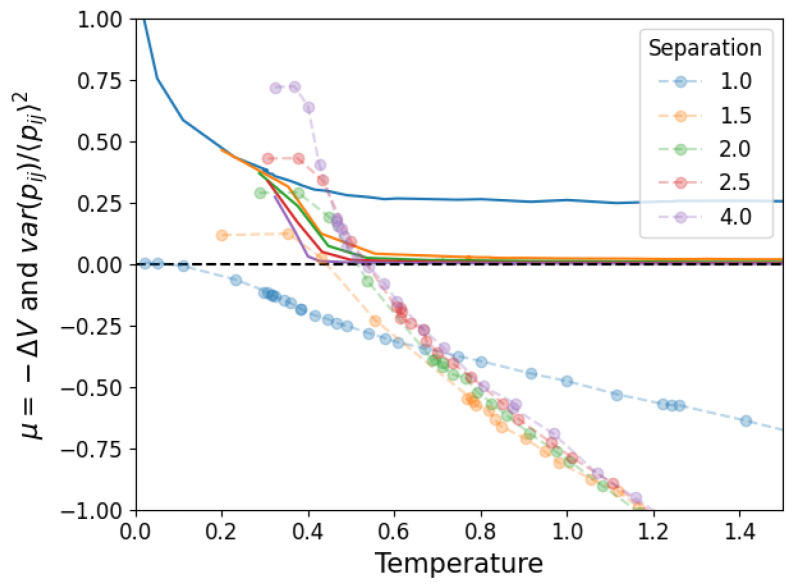
Relative fluctuations measured from simulations (in full lines) and chemical potential calculated from the corresponding data. The chemical potential predicts a transition from stable to unstable structure at a slightly higher temperature than observed.

## Data Availability

Python code for generating the simulation data and results, as well as the data themselves, is available on request from the corresponding author in order to comply with the data management policy of the Integrated Research on Energy, Environment and Society.

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
