# Peer review of "Complex Networks and Interacting Particle Systems"

_entropy, 2023, doi:10.3390/e25111490_

Round 1

Reviewer 1 Report

Comments and Suggestions for Authors The paper is dedicated to establishing an interesting connection between complex network theory and partition functions of classical statistical systems. I do find the approach interesting, and I am generally in favor of recommending the paper for publication. At the same time, I have a few comments   First of all, the writing style is not very sharp. While reading the paper, it is easy to get lost in wordy explanations of intermediate results, and the main message remains elusive until the very end. So, I recommend restructuring the narrative, making the main statements more explicit, and moving some of the plots to the Appendix.     Secondly, although the paper is named "A network perspective on structural complexity", there is no discussion of/connection with the existing discourse of structural complexity (e.g., thermodynamic depth complexity, hierarchical complexity, multi-scale structural complexity, and many others). Just computing a few characteristics of the network does not tell much about its _complexity_. Hence, the authors either need to connect their study with the theory of structural complexity in one way or another, or change the perspective and avoid making claims about complexity studies.   Finally, they need to discuss in more detail which potentials (apart from the considered Lennard-Jones) can be studied with this method. E.g., what about multi-valley potentials? Also, the authors say "It would also require further conditions on the relation between these three values (beyond the potential energy and the number of links). Assuming that this is possible, increasing the number of values to a countable infinity would produce a weighted network with integer valued levels, and finally the continuous limit continuous weights that match the potential." If feasible, this is indeed interesting. However, is it actually possible to say at least something about the relevant parameters at least for the case of three-valued potential? Which variables beyond the energy and the number of links will enter the game.  

Once the author improve the presentation style and better contextualize the paper, I will be happy to recommend it for publication.

Comments on the Quality of English Language

The English per se (grammar, vocabulary etc.) is fine, but I am concerned with the presentation style. Too wordy, and not really to-the-point.

Author Response

Dear reviewer,
we thank you deeply for the time and effort to read and provide constructive criticism on our work. We have followed comments and suggestions as closely as possible, incorporating them into the manuscript. In particular:

Regarding the writing style and the perspective: we have decided to take the advice of changing the perspective of the article to focus on the relation between networks and physical structures they may represent when composed of interacting particle systems, without invoking complexity. We have largely rewritten the manuscript, paying particular attention to the introduction, the last part of the results and the conclusions to make it clearer and more to the point.

Regarding other potentials and weighted networks: A short appendix (F) has been added on this point, along with a mention in the conclusions on how potentials are replaced with maximum entropy constraints in the general case where an interaction potential is not necessarily defined.

It should also be noted that an error in the relation between the average number of links and the chemical potential was found (missing a factor 1/2 with respect to the partition function). This error was dragged into the code and fixing it has resulted in a change in the relation between chemical potential and temperature (current figure 11), making the high density curve still different from the others but much less so than before.

With this, we hope to have fully addressed your comments and requests.

Kind regards and thank you,
Noam Abadi and Franco Ruzzenenti

Reviewer 2 Report

Comments and Suggestions for Authors

Author Response

Dear reviewer,
we thank you deeply for the time and effort to read and provide constructive criticism on our work. We have followed comments and suggestions as closely as possible, incorporating them into the manuscript. In particular, we have followed the advice of largely rewriting the manuscript, taking a perspective on the relation between complex network and physical structures composed of interacting particle systems and in particular, without invoking complexity. Efforts have focused on the introduction, the last part of the results and the conclusions. 

Regarding technical aspects:

    The Vo used for the description of the Lennard-Jones potential is not the same as used in the construction of the two-level approximation. We have changed the notation to Va and Vb throughout the document to avoid confusion.

    A short discussion on the choice of rT = 1 is now given in the second-to-last paragraph of section 2.2: Network construction.

    In the previous version of the manuscript, the comment on the functional form of the velocity distribution referred to the fact that if this is the partition function can be written in this way (essentially, no extra constraints), then the equipartition theorem tells us that the temperature is equal to <v dH/dv> and to <-q dH/dq>, in which the first equality is proportional to the kinetic energy regardless of the potential. This temperature is the same that goes into the partition function and, as the partition function is separable, determines the shape of the corresponding maxwell-boltzmann distribution. We have simplified this in the new version of the manuscript, referencing Kardar's statistical mechanics of particles, and shown the agreement between kinetic energy and temperature fitted from the maxwell-boltzmann distribution.

    As is now mentioned in the text, assortativity is a global measure, meaning that we only have one value per instant (instead of node specific values like for the degree and clustering), which worsens the statistic.

    We have added figures for two different approximations in which the values Va and Vb (former Vo and V1) can be calculated, showing agreement with the simulation and the imposed parameters. It should also be noted that an error in the relation between the average number of links and the chemical potential was found (missing a factor 1/2 with respect to the partition function). This error was dragged into the code and fixing it has resulted in a change in the relation between chemical potential and temperature (current figure 11), making the high density curve still different from the others but much less so than before.

    We have removed mentions of complexity and energy density, save one of each in the introduction and conclusions respectively.

With this, we hope to have fully addressed your comments and requests.

Kind regards and thank you,
Noam Abadi and Franco Ruzzenenti